# Gender-based violence and depressive symptoms among female entertainment workers in Cambodia: A cross-sectional study

**Sophearen Ith[1], Siyan Yi[2,3,4]\*, Sovannary Tuot[1,2], Sokunthea Yem[5], Pheak Chhoun[2], Masamine Jimba[1], Akira Shibanuma[1]**

**1** Department of Community and Global Health, Graduate School of Medicine, The University of Tokyo, Tokyo, Japan, **2** KHANA Center for Population Health Research, Phnom Penh, Cambodia, **3** Saw Swee Hock School of Public Health, National University of Singapore and National University Health System, Singapore, Singapore, **4** Center for Global Health Research, Touro University California, Vallejo, California, The United States of America, **5** National Institute for Public Health, Phnom Penh, Cambodia

\* siyan@doctor.com

**Data Availability Statement:** All data are in the Supporting information files.

## Abstract

Female entertainment workers (FEWs) are at higher risk of gender-based violence (GBV) than the general population. The prolonged stress and fear caused by GBV increase the likelihood of depression, a major mental health problem among FEWs. However, their mental health issue has received limited attention and remains poorly researched in the context of GBV. We examined the association between GBV and depressive symptoms among FEWs in Cambodia. We conducted this cross-sectional study in 2017. We used a two-stage cluster random sampling method to select FEWs from the municipality and six provinces for face-to-face interviews. We used the Centre for Epidemiologic Studies Depression Scale (CES-D) to measure depressive symptoms. We conducted a multivariable logistic regression analysis to identify factors associated with depressive symptoms. We included a total of 645 FEWs in data analyses. The proportions of FEWs experiencing emotional, physical, and sexual violence were 36.1%, 11.6%, and 17.2%, respectively. Of the total participants, 65.9% had high levels of depressive symptoms. The adjusted odds of having high levels of depressive symptoms were higher among FEWs who engaged in transactional sex (AOR 1.79, 95% CI 1.09–2.94), experienced emotional abuse (AOR 3.15, 95% CI 1.90–5.23), and experienced two (AOR 7.89, 95% CI 3.28–18.99) and three overlapping types of GBV (AOR 12.12, 95% CI 2.47–59.25) than those who did not. FEWs in this study experienced high levels and overlapping types of GBV associated with high levels of depressive symptoms. Policy interventions and services should be designed to prevent GBV and support the victims of GBV to mitigate depressive symptoms among FEWs in Cambodia.

## Introduction

Gender-based violence (GBV) is a global pandemic [1]. GBV is a harmful threat directed at any group or individual based on actual or perceived sex, gender identity, and expression, or

**Funding:** The project was funded by the United States Agency for International Development through the HIV/AIDS Flagship Project (Reference no. RFA-412-12-000003), with SY as the principal recipient and ST and PC as co-recipients. The funders had no role in study design, data collection and analysis, decision to publish, or preparation of the manuscript.

**Competing interests:** The authors have declared that no competing interests exist.

lack of adherence to various socially constructed norms [1]. The acts of GBV include physical and sexual violence, economic deprivation, threats, blackmail, and psychological abuse [1].

Culture, social class, and poverty are well-known determinants of GBV risk among women [2]. According to a systematic review of the correlates of violence against sex workers in 39 countries, female sex workers (FSWs), who often come from poor and rural families, are at high risk of GBV [3]. In Soweto, South Africa, FSWs reported significantly higher rates of GBV than the general population for reasons including social norms, increased exposure to violent partners, and lack of support services [4]. In the United States, FSWs experienced high physical and sexual violence from different perpetrators, including clients, intimate partners, police, and pimps [5]. Over 40% reported such abuses within the past month alone [5].

With or without physical injuries, which can be near-fatal or fatal, GBV often affects the mental health of the victims [6]. Prolonged stress and fear caused by GBV increase the likelihood of depression, one of the significant mental health problems among FSWs [6, 7]. According to a systematic review, a pooled prevalence of probable depression among FSWs was 62.4% [8]. In Nepal, 82.4% of FSWs had depressive symptoms, and FSWs who had experienced GBV were five times more likely to have depressive symptoms than those who had not [9]. Similarly, experiencing sexual violence was associated with high depressive symptoms among FSWs in China [10].

In Cambodia, female entertainment workers (FEWs) are at risk of GBV. They are defined as women or girls who work at entertainment venues, including restaurants, bars, nightclubs, karaoke, street-based, massage parlors, or beer gardens [11]. FEWs receive payment from clients for entertaining them at entertainment venues, including transactional sex [12, 13]. Not all FEWs exchange sex for money. However, a growing number of FEWs exchange sex for money with clients due to the implementation of the 2008 "Law on Suppression of Human Trafficking and Sexual Exploitation", which banned brothel-based sex work [14, 15]. In addition, the global economic crisis in 2007 has made a massive influx of unemployed women working in entertainment venues [16]. The number of FEWs in Cambodia almost doubled, from approximately 40,000 in 2014 to 70,000 in 2019 [17, 18].

In Cambodia, FEWs experience high levels and multiple forms of GBV confronted with different perpetrators, including intimate partners, clients, and entertainment establishment owners or managers [19]. Two-thirds of FEWs reported having experienced GBV, including emotional abuse, physical abuse, forced sex, and forced substance use in their lifetime [19]. FEWs, especially those who engage in transactional sex, face considerable risks to GBV, including rape and gang rape, by clients [13].

FEWs are at higher risk of mental health problems than general women because of poverty, the nature of their work, and lack of support service after experiencing GBV [20, 21]. They experience high psychological distress, which likely stems from current working conditions and past negative experiences, including having a history of emotional abuse or having a parent or guardian who was physically abused [12].

The poor mental health of FEWs can lead to more health risk behaviors such as substance use or unsafe sex. Nevertheless, their mental health has received limited attention and remains poorly researched in light of the intersecting epidemics of GBV [22]. More evidence is needed to develop effective intervention programs to narrow this gap with the commitment to "leaving no one behind" of the 2030 Agenda of the Sustainable Development Goals [23]. Therefore, we conducted this study to examine the relationship between different forms of GBV and depressive symptoms among FEWs in Cambodia.

## Materials and methods

### Ethics statement

We received ethical approval from the National Ethics Committee for Health Research, the Ministry of Health in Cambodia (N222NECHR) and the Research Ethics Committee of the University of Tokyo (2020026NI). Before starting the interviews, data collectors explained to FEWs about the study in the Khmer language and obtained written informed consent from them. Data collectors also provided information about free counseling and relevant referral services in case FEWs wanted to speak to someone during or after the study. We conducted the interviews in a private location. We assigned a code number for each FEW to ensure anonymity and confidentiality. We also emphasized confidentiality and identity protection during the data collection training. Each FEW received USD 5 for their transport and time compensation.

### Study design

We conducted this cross-sectional study in October 2017.

### Setting

We recruited FEWs from a cluster of entertainment venues from seven sites with the highest number of FEWs and entertainment venues in Cambodia [24]. Those seven sites were the capital city of Phnom Penh and six provinces, including Banteay Meanchey, Battambang, Kampong Cham, Preah Sihanouk, Pursat, and Siem Reap [24]. The total number of FEWs in these seven sites was about 50% of the total FEWs in Cambodia in 2017 [24].

### Participants

We included a woman in this study if she: (1) was 18 years or older, (2) self-identified as a FEW, (3) identified herself as a cis-woman (born female and self-identifies as a woman) (we did not include transgender women because they experienced higher GBV and depressive symptoms) [25], (4) could communicate in Khmer, the national language of Cambodia, and (5) had lived within the seven selected sites for at least three months before the data collection.

### Sample size calculation

We calculated the sample size based on the objective of the main study, which is to estimate the prevalence of GBV among FEWs [26]. The total number of FEWs in the seven sites was 20,182 in 2017 [24]. The estimated prevalence of GBV among FEWs was unknown when we designed the study; therefore, we used 50% for the sample size calculation to prevent any underestimated prevalence [27]. Based on a 95% confidence interval (CI), a +5% margin of error, and a 1.5 design effect, the minimum required sample size was 652 FEWs, with 10% adjusted for incomplete responses, missing data, and refusal rate.

### Sampling

We used a two-stage cluster random sampling method to select entertainment venues. First, we obtained a list of venues in the seven study sites from the program team of the Khmer HIV/AIDS NGO Alliance (KHANA), a local non-government organization. We excluded venues with less than three FEWs for logistic efficiency and listed the remaining 803 venues across the study sites. Second, we randomly selected 10% of the total number of entertainment venues in each study site. However, to ensure a broader representation of women, we included a

minimum of ten venues if a site had less than 100 venues. We randomly selected a maximum of seven FEWs from each included venue. For venues having seven or fewer FEWs, we applied a take-all approach. Following adjustments, we included 102 entertainment venues with 652 FEWs in the study. We generated a reserved list of venues and FEWs and used them as alternates to complete the same size if the number of FEWs in the selected venues could not reach the minimum required sample size. We then contacted the venues' managers and outreach workers to get a list of FEWs in each venue.

## Data collection training and procedure

Experienced and well-trained female data collectors conducted data collection. All data collectors and field supervisors received three days of data collection training. The training covered confidentiality, privacy, and interview techniques and provided opportunities for the study teams to rehearse questionnaire administration and other study procedures. We conducted a tool pre-test with nine FEWs at one entertainment venue in Phnom Penh. After the pre-test, we refined the questionnaire in a reflection meeting. We excluded FEWs who participated in the pre-test from the main study. On the data collection day, we randomly recruited FEWs from a name list of FEWs obtained from each selected entertainment venue's manager and outreach worker, as described above. Next, the data collectors contacted the selected FEWs by telephone. Then, data collectors screened the selected FEWs for eligibility with a screening tool and made an appointment with them for a face-to-face interview. The data collectors obtained written consent from FEWs. The data collectors recorded responses on paper-based questionnaires. We transported the questionnaires to Phnom Penh weekly for double-data entry using EpiData 3.1. The project coordinator oversaw and verified data completion and correctness. The data collectors conducted review sessions regularly to review progress and communicate any problems during data collection.

## Questionnaire development

The structured questionnaire included socio-demographic characteristics, entertainment work history, sex work, experiences of GBV, and depressive symptoms. Initially, we established the questionnaire in English and translated it into Khmer. Then, another translator translated it back into English to sustain the content and meaning of every original item. Finally, we included clear instructions and explanations to avoid confusion during interviews.

## Variables and measurements

**Depressive symptoms.** We used the Center for Epidemiologic Studies Depression Scale (CES-D) to measure depressive symptoms [28]. The validity of the CES-D scale has been well established in Western [28] and Asian [29] populations. This scale has been used in many studies in Cambodia [25, 30]. This scale consists of 20 questions covering six symptoms of depression: guilt or worthlessness, depressed mood, helplessness or hopelessness, loss of appetite, psychomotor retardation, and sleep disturbance experienced during the past seven days. Four items (I felt I was just as good as other people, I felt hopeful about the future, I was happy, I enjoyed life) were worded in a positive direction. We reversed coded these four items in the analyses. The response options of each question are on a scale of zero to three according to the frequency of occurrence of the symptoms [0 = Never, 1 = Occasionally (1-2days), 2 = Sometime (3–4 days), 3 = Almost all the time (5–7 days)]. The total CES-D score ranges from 0 to 60. We defined a FEW with a total CES-D score of <16 as having low levels of depressive symptoms and a FEW with a total CES-D score of ≥16 as having high levels of depressive symptoms [28]. This threshold aids in identifying individuals at risk for clinical

depression with good sensitivity and specificity and high internal consistency [31]. The Cronbach's alpha of the CES-D scale in this study is 0.87.

**Experience of GBV.** We adapted questionnaires from the WHO Multi-Country Study on Women's Health and Domestic Violence against Women, following the Guidelines for Producing Statistics on Violence against Women [27, 32]. We also included questions specific to this context gathered from initial meetings with FEWs. We asked FEWs about their experiences in different forms of GBV in the past 12 months, including physical violence (ever been hit or thrown something, been threatened with a gun or weapon, been choked or burned on purpose), sexual violence (been forced to do something sexually, not been paid for sex or paid less, been forced to do something sexual, been forced to engage in group sex), and emotional abuse (been yelled at or insulted, been intimidated or scared and threatened with harm). We dichotomized the response into yes or no to determine the proportions of ever experiencing at least one form of GBV described above in the past 12 months.

**Socio-demographic characteristics, entertainment work, and sex work.** We used items from the most recent Cambodia Demographic and Health Survey in 2014 [33] and previous studies in Cambodia [12, 34]. Socio-demographic characteristics included study site, age, marital status, the average monthly income in the last three months, number of dependents, perceived family economic status when they were growing up, completed years of formal education, living companions, and living arrangement. The entertainment work information included duration of work as FEWs, having other jobs, having borrowed money in the last three months, type of entertainment workplace, and satisfaction with the current job. We also asked whether they have ever been forced to drink alcohol at work. For transactional sex, we asked whether they have ever exchanged sex for money, goods, or other benefits in the past three months.

## Data analyses

We used STATA version 13 (College Station, Texas, USA) for statistical analyses. We excluded FEWs whose information was partly missing. We performed descriptive analyses of socio-demographic characteristics, entertainment work, sex work, depressive symptoms, and the experience of GBV. In bivariate analyses, we compared socio-demographic characteristics, entertainment work, sex work, and GBV experiences among FEWs with low and high depressive symptoms. For categorical variables, we used the Chi-square test or Fisher's exact test if the number of FEWs was smaller than five in at least one cell. For continuous variables, we used Student's t-test for the significance testing.

We conducted a multilevel logistic regression analysis with a random intercept at the study site and entertainment venue levels to examine if the outcome variable had extensive cluster effects at the site and entertainment venue levels. Upon reviewing the cluster effects using the multilevel logistic regression model with random intercepts at the site (SD 0.09) and entertainment place levels (SD 0.20), the standard deviations of the random effect variables were small enough to reject the existence of the cluster effects. Therefore, we used the multivariable logistic regression model to examine the association between GBV and depressive symptoms, controlling for potential confounders. We constructed two separate models. In the first model, we included three types of GBV (emotional abuse, physical, and sexual violence). In the second model, we included variables representing overlapping types of GBV (never experienced, experienced one type, experienced two types, and experienced three types). We included socio-demographic characteristics, entertainment work, and sex work in both models, following the literature [13, 17, 24]. We set the significance level at 5%. No multicollinearity was detected. We obtained the adjusted odds ratios (AORs) and presented them with confidence intervals (CIs) and $p$-values.

## Results

### Socio-demographic characteristics

This study included 645 FEWs after excluding seven FEWs who missed answering CES-D questions. As shown in Table 1, 47.6% of FEWs were 18–25 years old, and 65.9% had high levels of depressive symptoms. Of the total, 44.5% were widowed/separated/divorced, with an average monthly income of 260 USD (SD 150).

### Entertainment and sex work

Table 2 shows that 65.4% of the FEWs had worked as FEWs for less than 17 months. The proportions of FEWs working in karaoke, massage parlors, and restaurants were 43.1%, 16.9%, and 18.5%, respectively. More than two-thirds (70.6%) were mostly or very satisfied with their job. About one-third (32.7%) engaged in transactional sex with clients for money, goods, or other benefits in the last three months.

### Gender-based violence (GBV)

Table 3 presents the exposure to different forms of GBV among FEWs. Of the total, 41.9% had experienced at least one type of GBV, and 18.0% had experienced overlapping types of GBV in the last 12 months. The proportions of FEWs experiencing emotional, physical, and sexual violence were 36.1%, 11.6%, and 17.2%, respectively. In bivariate analyses, experiencing emotional ($p<0.001$), physical ($p<0.001$), and sexual violence ($p<0.001$) were significantly associated with depressive symptoms.

### Factors associated with depressive symptoms

Table 4 presents multivariable logistic regression analysis results of factors associated with depressive symptoms among FEWs. After controlling for other covariates in the model, the adjusted odds of having high levels of depressive symptoms were significantly higher among FEWs who engaged in transactional sex (AOR 1.79, 95% CI 1.09–2.94) and experienced emotional abuse in the past 12 months (AOR 3.15, 95% CI 1.90–5.23).

Table 5 shows multivariable logistic regression results of the association between experiencing overlapping types of GBV and depressive symptoms, including the same set of covariates in Table 4. The adjusted odds of having high levels of depressive symptoms were significantly higher among FEWs who experienced one type (AOR 1.90, 95% CI 1.18–3.06), two types (AOR 7.89, 95% CI 3.28–18.99), and three types (AOR 12.12, 95% CI 2.47–59.25) than those who had never experienced GBV.

## Discussion

This study has two key findings. First, among the three types of GBV, emotional abuse was the most experienced by FEWs and independently associated with depressive symptoms. Second, FEWs were more likely to have high levels of depressive symptoms if they had experienced more overlapping types of GBV among emotional, physical, and sexual violence.

Among three types of GBV, emotional abuse was the most experienced by FEWs in this study. Emotional abuse was also associated with depressive symptoms among FEWs. This result is similar to other related studies investigating the association between emotional abuse and mental health among FSWs [35, 36]. However, the settings of these studies were different as not all FEWs in our study engaged in transactional sex like FSWs, and mental health was measured differently [35, 36]. In Cambodia, men tend to exercise their presumed male power through verbal threatening (rather than beating) to reinforce or reinstate their dominance

**Table 1. Comparison of socio-demographic characteristics of FEWs with low and high levels of depressive symptoms (*n* = 645).**

| Variable | Total (*n* = 645) | Depressive symptoms* | | *p*-value† |
| --- | --- | --- | --- | --- |
| | | Low (*n* = 220) | High (*n* = 425) | |
| Study site | | | | 0.04 |
| Phnom Penh | 267 (41.4) | 92 (34.5) | 175 (65.5) | |
| Pursat | 70 (10.9) | 22 (31.4) | 48 (68.6) | |
| Battambang | 70 (10.8) | 25 (35.7) | 45 (64.3) | |
| Banteay Meanchey | 63 (9.8) | 25 (39.7) | 38 (60.3) | |
| Siem Reap | 73 (11.3) | 13 (17.8) | 60 (82.2) | |
| Kampong Cham | 49 (7.6) | 21 (42.9) | 28 (57.1) | |
| Preah Sihanouk | 53 (8.2) | 22 (41.5) | 31 (58.5) | |
| Age group | | | | 0.03 |
| 18–25 | 307 (47.6) | 111 (36.2) | 196 (63.8) | |
| 26–34 | 272 (42.2) | 96 (35.3) | 176 (64.7) | |
| 35–48 | 66 (10.2) | 13 (19.7) | 53 (80.3) | |
| Marital status | | | | <0.001 |
| Single | 225 (34.9) | 98 (43.6) | 127 (56.4) | |
| Married | 133 (20.6) | 47 (35.3) | 86 (64.7) | |
| Widowed/separated/divorced | 287 (44.5) | 75 (26.1) | 212 (73.9) | |
| Monthly income (x US$100) | 2.6 (1.5) | 2.6 (1.3) | 2.61 (1.5) | 0.74 |
| Number of dependents | | | | 0.05 |
| No dependent | 36 (5.6) | 15 (41.7) | 21 (58.3) | |
| 1 dependent | 91 (14.1) | 39 (42.9) | 52 (57.1) | |
| 2 dependents | 130 (20.2) | 49 (37.7) | 81 (62.3) | |
| 3+ dependents | 388 (60.2) | 117 (30.2) | 271 (69.8) | |
| Childhood family economic status | | | | 0.04 |
| Poor | 370 (57.4) | 113 (30.5) | 257 (69.5) | |
| Medium | 253 (39.2) | 101 (39.9) | 152 (60.1) | |
| Rich | 22 (3.4) | 6 (27.3) | 16 (72.7) | |
| Education | | | | 0.57 |
| No education | 60 (9.3) | 20 (33.3) | 40 (66.7) | |
| Primary | 309 (47.9) | 98 (31.7) | 211 (68.3) | |
| Secondary | 215 (33.3) | 78 (36.3) | 137 (63.7) | |
| High school | 61 (9.5) | 24 (39.3) | 37 (60.6) | |
| Living companions | | | | 0.63 |
| Alone | 97 (15.0) | 29 (29.9) | 68 (70.1) | |
| Relatives | 219 (33.9) | 81 (37.0) | 138 (63.0) | |
| Friends | 221 (34.3) | 75 (33.9) | 146 (66.1) | |
| Sexual partner/husband | 108 (16.7) | 35 (32.4) | 73 (67.6) | |
| Living condition | | | | 0.37 |
| Renting room | 355 (55.1) | 113 (31.8) | 242 (68.2) | |
| Own house | 42 (6.5) | 13 (30.9) | 29 (69.1) | |
| Friend's house | 46 (7.1) | 15 (32.6) | 31 (67.4) | |
| Relative's house | 49 (7.6) | 22 (44.9) | 27 (55.1) | |

(*Continued*)

**Table 1.** (Continued)

| Variable | Total | Depressive symptoms* | | |
|---|---|---|---|---|
| | (*n* = 645) | Low (*n* = 220) | High (*n* = 425) | *p*-value† |
| Workplace | 153 (23.7) | 57 (37.2) | 96 (62.8) | |

FEWs, female entertainment workers.

Values are the number of subjects (%) for categorical variables and the mean (standard deviation) for continuous variables.

*Measured by a Center for Epidemiology Studies Depression Scale (CES-D). A total CES-D score of 16 was used as a cut-off; ≥16: high levels of depressive symptoms, <16: low levels of depressive symptoms.

†Chi-square test, or Fisher's exact test when the sample sizes were smaller than five in one cell, was used for categorical variables. Independent Student's t-test was used for continuous variables.

**Table 2. Comparison of entertainment and sex work of FEWs with low and high levels of depressive symptoms (*n* = 645).**

| Variable | Total | Depressive symptoms* | | |
|---|---|---|---|---|
| | (*n* = 645) | Low | High | *p*-value† |
| | | (*n* = 220) | (*n* = 425) | |
| Duration of entertainment work (<17 months)‡ | 422 (65.4) | 138 (32.7) | 284 (67.3) | 0.30 |
| Had borrowed any money in the last 3 months | | | | <0.001 |
| No | 374 (58.0) | 159 (42.5) | 215 (57.5) | |
| Yes | 271 (42.0) | 61 (22.5) | 210 (77.5) | |
| Type of current working place | | | | 0.35 |
| Karaoke | 278 (43.1) | 92 (33.1) | 186 (66.9) | |
| Massage parlor | 109 (16.9) | 39 (35.8) | 70 (64.2) | |
| Beer garden | 26 (4.0) | 4 (15.4) | 22 (84.6) | |
| Restaurant | 119 (18.5) | 42 (35.3) | 77 (64.7) | |
| Dance club | 82 (12.7) | 30 (36.6) | 52 (63.4) | |
| Street | 10 (1.5) | 3 (30.0) | 7 (70.0) | |
| Bar | 21 (3.3) | 10 (47.6) | 11 (52.4) | |
| Satisfaction with the current job | | | | <0.001 |
| Unsatisfied | 21 (3.3) | 3 (14.3) | 18 (85.7) | |
| Somewhat satisfied | 143 (22.2) | 24 (16.8) | 119 (83.2) | |
| Neutral | 26 (4.0) | 4 (15.4) | 22 (84.6) | |
| Mostly satisfied | 399 (61.9) | 165 (41.3) | 234 (58.7) | |
| Very satisfied | 56 (8.7) | 24 (42.9) | 32 (57.1) | |
| Ever been forced to drink | | | | 0.001 |
| No | 547 (84.8) | 201 (36.7) | 346 (63.3) | |
| Yes | 98 (15.2) | 19 (19.4) | 79 (80.6) | |
| Engage in transactional sex | | | | <0.001 |
| No | 434 (67.3) | 172 (39.6) | 262 (60.4) | |
| Yes | 211 (32.7) | 48 (22.7) | 163 (77.3) | |

FEWs, female entertainment workers.

Values are the number of subjects (%).

*Measured by a Center for Epidemiology Studies Depression Scale (CES-D). A total CES-D score of 16 was used as a cut-off; ≥16: high levels of depressive symptoms, <16: low levels of depressive symptoms.

†Chi-square test, or Fisher's exact test when the sample sizes were smaller than five in one cell, was used.

‡Mean value was used to categorize FEWs.

**Table 3. Comparison of past year experience of GBV among FEWs with low and high levels of depressive symptoms (n = 645).**

| Variable | Total (*n* = 645) | Depressive symptoms* | | |
| --- | --- | --- | --- | --- |
| | | Low (*n* = 220) | High (*n* = 425) | *p*-value† |
| Overlap experience of violence | | | | <0.001 |
| Never experienced violence | 375 (58.1) | 163 (43.5) | 212 (56.5) | |
| Experienced one type | 154 (23.9) | 48 (31.2) | 106 (68.8) | |
| Experienced two types | 83 (12.9) | 7 (8.4) | 76 (97.6) | |
| Experienced three types | 33 (5.1) | 2 (6.1) | 31 (93.9) | |
| Physical violence‡ | 75 (11.6) | 7 (9.3) | 68 (90.7) | <0.001 |
| Hit, beaten, slapped, kicked | 61 (9.5) | 6 (9.8) | 55 (90.2) | <0.001 |
| Threatened with a gun, knife | 26 (4.0) | 2 (7.7) | 24 (92.3) | 0.003 |
| Sexual violence‡ | 111 (17.2) | 18 (16.2) | 93 (83.8) | <0.001 |
| Forced to do sexual acts when you don't want | 47 (7.3) | 8 (17.0) | 39 (83.0) | 0.01 |
| Not been paid for sex | 77 (11.9) | 14 (18.2) | 63 (81.8) | 0.002 |
| Forced to do humiliating sexual behavior | 9 (1.4) | 0 (0.0) | 9 (100) | 0.03 |
| Forced to engage in group sex | 13 (2.0) | 1 (7.7) | 12 (92.3) | 0.07 |
| Emotional violence‡ | 233 (36.1) | 43 (18.4) | 190 (81.6) | <0.001 |
| Yelled at, insulted, or belittled | 224 (34.7) | 42 (18.7) | 182 (81.3) | <0.001 |
| Intimidated or scared on purpose and threatened with harm | 47 (7.3) | 5 (10.6) | 42 (89.4) | <0.001 |

FEWs, female entertainment workers; GBV, gender-based violence.

Values are the number of subjects (%).

*Measured by a Center for Epidemiology Studies Depression Scale (CES-D). A total CES-D score of 16 was used as a cut-off; ≥16: high levels of depressive symptoms, <16: low levels of depressive symptoms.

†Chi-square test, or Fisher's exact test when the sample sizes were smaller than five in one cell, was used.

‡When FEWs experienced at least one of the below variables.

[37]. Emotional abuse is widespread among FEWs as their work environment promotes and allows violent behavior from the perpetrators [19]. FEWs often experience these emotional abuses from the entertainment venue owners or managers, partners/husbands, co-workers, community members, and most frequently from the clients when they are drunk [13, 19]. Emotional abuse is a form of GBV that directly impairs self-esteem, especially intimidation in front of others [38]. However, emotional abuse is less visible to society and can lead to physical violence [37]. This finding is essential as emotional abuse is usually considered a minor type of GBV and receives less attention from the victims, legal supporters, policymakers, medical staff, and researchers [39].

The experience of overlapping types of GBV was independently associated with depressive symptoms among FEWs in this study. The odds of having high depressive symptoms progressively increased as FEWs experienced more overlapping types of GBV in this study. Traumatic and psychological stress reactions are the core mechanisms that explain why GBV may cause subsequent depression [6]. Physical violence combined with emotional abuse can result in feelings of frustration and motivational impairment, including low self-esteem and passivity, leading to depression [38]. In the regression model that examined the separate role of each GBV type, physical and sexual violence had no association with depressive symptoms among FEWs. This result contrasts with a study conducted among FSWs in Mongolia, where sexual and physical violence was associated with depressive symptoms [7]. However, the results of this study do not necessarily mean that sexual and physical violence was less critical.

**Table 4. Factors associated with depressive symptoms among FEWs (*n* = 645).**

| Variable in the model | Depressive symptoms* | |
|---|---|---|
| | AOR (95% CI) | *p*-value |
| Study site | | |
| Phnom Penh | Reference | |
| Pursat | 1.12 (0.55–2.28) | 0.74 |
| Battambang | 1.46 (0.73–2.92) | 0.28 |
| Banteay Meanchey | 0.56 (0.26–1.20) | 0.14 |
| Siem Reap | 1.38 (0.61–3.11) | 0.42 |
| Kampong Cham | 1.23 (0.57–2.66) | 0.59 |
| Preah Sihanouk | 1.10 (0.44–2.71) | 0.82 |
| Age group | | |
| 26–34 | Reference | |
| 18–25 | 1.76 (1.10–2.83) | 0.01 |
| 35–48 | 3.01 (1.39–6.52) | 0.005 |
| Marital status | | |
| Single | Reference | |
| Married | 0.73 (0.34–1.60) | 0.44 |
| Widow/separate/divorce | 1.75 (1.05–2.92) | 0.02 |
| Monthly income | 0.95 (0.82–1.10) | 0.53 |
| Number of dependents | | |
| No dependent | Reference | |
| 1 dependent | 1.17 (0.46–3.00) | 0.72 |
| 2 dependents | 1.65 (0.67–4.07) | 0.27 |
| 3+ dependents | 1.86 (0.79–4.37) | 0.14 |
| Childhood family economic status | | |
| Poor | Reference | |
| Medium | 0.91 (0.60–1.37) | 0.66 |
| Rich | 1.01 (0.33–3.08) | 0.97 |
| Education | | |
| No education | Reference | |
| Primary | 1.31 (0.63–2.72) | 0.47 |
| Secondary | 1.01 (0.46–2.18) | 0.97 |
| High school | 0.97 (0.37–2.53) | 0.95 |
| Living companions | | |
| Alone | Reference | |
| Relatives | 0.76 (0.40–1.43) | 0.40 |
| Friends | 1.04 (0.55–1.97) | 0.88 |
| Sexual partner/husband | 1.45 (0.57–3.71) | 0.43 |
| Duration of working as a FEW† | | |
| < 17 months | Reference | |
| ≥ 17 months | 0.84 (0.55–1.29) | 0.44 |
| Had borrowed money from someone in the last 3 months | | |
| No | Reference | |
| Yes | 1.89 (1.25–2.88) | 0.003 |
| Type of current working place | | |
| Karaoke | Reference | |
| Massage parlor | 1.18 (0.67–2.06) | 0.55 |
| Beer garden | 2.35 (0.63–8.69) | 0.20 |

(*Continued*)

**Table 4.** (Continued)

| Variable in the model | Depressive symptoms* | |
|---|---|---|
| | AOR (95% CI) | *p*-value |
| Restaurant | 1.32 (0.76–2.29) | 0.32 |
| Dance club | 0.59 (0.29–1.20) | 0.15 |
| Street | 0.34 (0.06–1.93) | 0.22 |
| Bar | 0.44 (0.12–1.62) | 0.22 |
| Current job satisfaction | | |
| Unsatisfied | Reference | |
| Somewhat satisfied | 1.49 (0.34–6.46) | 0.59 |
| Neutral | 1.95 (0.32–11.64) | 0.46 |
| Mostly satisfied | 0.42 (0.10–1.73) | 0.23 |
| Very satisfied | 0.36 (0.07–1.68) | 0.19 |
| Ever been forced to drink | | |
| No | Reference | |
| Yes | 1.07 (0.56–2.03) | 0.82 |
| Engage in transactional sex in the last three months | | |
| No | Reference | |
| Yes | 1.79 (1.09–2.94) | 0.01 |
| Physical violence | | |
| No | Reference | |
| Yes | 2.36 (0.96–5.82) | 0.06 |
| Sexual violence | | |
| No | Reference | |
| Yes | 1.42 (0.73–2.77) | 0.29 |
| Emotional violence | | |
| No | Reference | |
| Yes | 3.15 (1.90–5.23) | <0.001 |

FEWs, female entertainment workers; AOR, adjusted odds ratio; CI, confidence interval.

*Measured by a Center for Epidemiology Studies Depression Scale (CES-D). A total CES-D score of 16 was used as a cut-off; ≥16: high levels of depressive symptoms, <16: low levels of depressive symptoms.

†Mean value was used to categorize FEWs.

**Table 5. Multivariable logistic regression between overlapping types of GBV and depressive symptoms among FEWs (*n* = 645).**

| Variables in the model* | Depressive symptoms† | |
|---|---|---|
| | AOR (95%CI) | *p*-value |
| Overlap experience of GBV | | |
| Never experienced GBV | Reference | |
| Experienced one type | 1.90 (1.18–3.06) | 0.008 |
| Experienced two types | 7.89 (3.28–18.99) | <0.001 |
| Experienced three types | 12.12 (2.47–59.25) | 0.002 |

GBV, gender-based violence; FEWs, female entertainment workers; AOR, adjusted odds ratio; CI, confidence interval.

*The same variables in Table 4 were included in the model except for emotional, physical, and sexual violence.

†Measured by a Center for Epidemiology Studies Depression Scale (CES-D). A total CES-D score of 16 was used as a cut-off; ≥16: high levels of depressive symptoms, <16: low levels of depressive symptoms.

FEWs in this study experienced a high rate of GBV, of whom almost half had experienced overlapping types of GBV in the last 12 months. Consistent with other studies, this result suggests that GBV often happens in multiple forms, as physical and sexual violence usually come together with emotional abuse [40, 41]. Studies in Mongolia and Kenya reported similar findings, in which FSWs reported experiencing a high prevalence and overlapping types of GBV [42, 43]. GBV and harassment commonly occur within the context of FEWs' work when the clients are drunk or do not want to use condoms [16]. FEWs, particularly those who engage in transactional sex, often experience rape or gang rape, coercion for condomless sex, prolonged/rough sex, and underpayment [13]. For instance, sexual violence occurs not just from forced sex but physical assault during the negotiation process [13]. FEWs are often beaten, pulled by the hair, and raped when they request the client to use a condom [13]. While we found no independent association between physical or sexual violence and depressive symptoms, the high degree of overlap between physical, sexual, and emotional violence may make identifying an independent association complex in the current investigation.

This study also showed that engaging in transactional sex was independently associated with depressive symptoms. This finding was in line with the previous studies [7, 41, 44]. This could be explained by the fact that engaging in transactional sex is not their willingness [15]. The main reasons for selling sex included not finding another job that provides them sufficient income, the need to give financial support to families, and paying the debts of family members [16, 45, 46]. Moreover, FEWs might have experienced adverse life events before engaging in transactional sex. Many FEWs engage in transactional sex after having their first non-consensual sex between the age of 14 and 18 [16]. Some of them having their virginity bought stemmed from familial financial constraints or being raped by a boyfriend, stepfather, or stranger [16].

In this study, most of the FEWs in this study were satisfied with their current work. This could be because they accept certain aspects of their work, including tips, dancing, singing, and feeling cared for by the entertainment establishment owners [16]. A previous study also showed that FEWs enjoyed their work compared to potential alternative choices, such as moving back to their village and cultivating rice like their parents [16]. Nevertheless, this result might not imply that the working environment in entertainment venues was favorable. Some FEWs were unsatisfied with their job, which might be due to the long working hours, the nature of the work, and their status in society [16].

Previous programs in Cambodia, such as the NGO-led SMARTGirl, provided a support network for GBV survivors [13]. However, these programs do not necessarily contain legal support to prevent GBV and emotional support from health professionals [13]. Moreover, FEWs often face difficulties accessing services due to societal stigmatization [16]. Addressing GBV and mental health problems among FEWs requires a multifaceted approach. These could be done by providing legal and emotional support that FEWs can easily access, promoting and raising awareness of GBV and mental health among FEWs, and involving entertainment establishment owners and managers in innovative interventions to address emotional, physical, and sexual violence.

## Strengths and limitations

This study possesses several strengths. First, unlike many previous studies that used a small sample size from a particular type of setting, this study had broader coverage of FEWs throughout the nation. The sampling covered seven major sites with the highest number of FEWs and entertainment venues in Cambodia. Second, this study obtained detailed information about GBV and specific experiences in each type of GBV. Third, this is the first study investigating different types of GBV and factors associated with depressive symptoms among FEWs.

However, some limitations need to be acknowledged. First, due to the cross-sectional design, we cannot establish the direction of the association between GBV and depressive symptoms. For example, women exposed to GBV are at increased risk of depression, and women who reported depressive symptoms are more likely to experience violence afterward [46–48]. Second, all measures, including the experiences of GBV and depressive symptoms, were self-reported, which may lead to under-reporting and over-reporting. However, to reduce these biases, the measures taken included employing female data collectors, back-translation of the questionnaire, pre-test before the data collection, and pre-appointment with FEWs.

## Conclusion

In conclusion, FEWs in this study experienced high rates of GBV and depressive symptoms. Policy interventions and services should be designed to prevent GBV and support the victims of GBV to mitigate depressive symptoms among FEWs in Cambodia.

## Supporting information

**S1 Table. Comparison of socio-demographic characteristics of FEWs with and without experiencing each type of GBV and at least one type of GBV (n = 645).**
(DOCX)

**S2 Table. Comparison of entertainment and sex work of FEWs with and without experiencing each type of GBV (n = 645).**
(DOCX)

**S1 Data.**
(XLSX)

## Acknowledgments

This study was funded by the United States Agency for International Development through the "HIV/AIDS Flagship Project" and implemented by KHANA Center for Population Health Research. The authors thank all data collectors and all participants for their contribution to this study.

## Author Contributions

**Conceptualization:** Sophearen Ith, Siyan Yi, Sovannary Tuot, Masamine Jimba, Akira Shibanuma.

**Data curation:** Sophearen Ith, Siyan Yi, Sokunthea Yem, Pheak Chhoun.

**Formal analysis:** Sophearen Ith, Siyan Yi, Akira Shibanuma.

**Funding acquisition:** Siyan Yi, Sovannary Tuot.

**Investigation:** Sophearen Ith, Siyan Yi, Sovannary Tuot, Sokunthea Yem, Pheak Chhoun, Masamine Jimba.

**Methodology:** Sophearen Ith, Siyan Yi, Sokunthea Yem, Pheak Chhoun, Masamine Jimba, Akira Shibanuma.

**Project administration:** Sovannary Tuot, Sokunthea Yem, Pheak Chhoun.

**Resources:** Siyan Yi, Sovannary Tuot.

**Software:** Sophearen Ith.

**Supervision:** Siyan Yi, Sovannary Tuot, Sokunthea Yem, Pheak Chhoun, Masamine Jimba, Akira Shibanuma.

**Validation:** Sophearen Ith, Siyan Yi, Masamine Jimba, Akira Shibanuma.

**Visualization:** Sophearen Ith, Siyan Yi, Masamine Jimba, Akira Shibanuma.

**Writing – original draft:** Sophearen Ith.

**Writing – review & editing:** Sophearen Ith, Siyan Yi, Sovannary Tuot, Sokunthea Yem, Pheak Chhoun, Masamine Jimba, Akira Shibanuma.

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
