## [Decision Letter · Decision Letter 0]

10 Jun 2022

PGPH-D-21-00861

Gender-based violence and depressive symptoms among female entertainment workers in Cambodia: a cross-sectional study

Dear Dr. Yi,

Thank you for submitting your manuscript to PLOS Global Public Health. After careful consideration, we feel that it has merit but does not fully meet PLOS Global Public Health’s publication criteria as it currently stands. Therefore, we invite you to submit a revised version of the manuscript that addresses the points raised during the review process.

We look forward to receiving your revised manuscript.

Kind regards,

Muthusamy Sivakami

Academic Editor

Journal Requirements:

1. Please send a completed 'Competing Interests' statement, including any COIs declared by your co-authors. If you have no competing interests to declare, please state "The authors have declared that no competing interests exist". 

a. Please clarify all sources of funding (financial or material support) for your study. List the grants (with grant number) or organizations (with url) that supported your study, including funding received from your institution. 

b. State the initials, alongside each funding source, of each author to receive each grant.

c. State what role the funders took in the study. If the funders had no role in your study, please state: “The funders had no role in study design, data collection and analysis, decision to publish, or preparation of the manuscript.”

3. In the online submission form you indicate that your data is not available for proprietary reasons and have provided a contact point for accessing this data. Please note that your current contact point is a co-author on this manuscript. According to our Data Policy, the contact point must not be an author on the manuscript and must be a third party. Please revise your data statement to a non-author institutional point of contact, such as a data access or ethics committee, and send this to us via return email. Please also include contact information for the third party organization, and please include the full citation of where the data can be found.

Additional Editor Comments (if provided):

The paper has a great potential to add value to one of the understudied area, GBV and mental health. Our reviewers are very positive about the paper. However, they have raised some important issues. You are requested to address them which will help us proceed with next step in our process.

Reviewers' comments:

Reviewer's Responses to Questions

**Comments to the Author**

1. Does this manuscript meet PLOS Global Public Health’s publication criteria? Is the manuscript technically sound, and do the data support the conclusions? The manuscript must describe methodologically and ethically rigorous research with conclusions that are appropriately drawn based on the data presented.

Reviewer #1: Yes

Reviewer #2: Yes

2. Has the statistical analysis been performed appropriately and rigorously?

Reviewer #1: Yes

Reviewer #2: Yes

3. Have the authors made all data underlying the findings in their manuscript fully available (please refer to the Data Availability Statement at the start of the manuscript PDF file)?

Reviewer #1: No

Reviewer #2: Yes

4. Is the manuscript presented in an intelligible fashion and written in standard English?

Reviewer #1: Yes

Reviewer #2: Yes

5. Review Comments to the Author

Reviewer #1: Introduction

Lines 49-50: "In particular, female sex workers (FSWs), who often come from poor and rural families, are at higher risk of GBV (3)." I think you need to add "in Cambodia" to this sentence. This is not generalisable at the global level.

Methods

Line 123: Please define KHANA

Lines 172-175: Were these thresholds based on previous research? Please explain.

Lines 178-179: The WHO study is related to Intimate Partner Violence - has it been adapted to allow measurement of non-partner violence in the context of this questionnaire? Or did the researchers include the WHO non-partner measure? More detail is required on the measurement tools.

Lines 181-183: The measure is described as "past 12 months" (line 181) and "ever" (line 183), it can not be both. Please clarify.

Line 223-230: While the researchers describe the ethical considerations in this section, there is limited description of safety considerations for the participants. I find this concerning when it appears that the study was undertaken with the knowledge of entertainment venue managers and managers may be perpetrators of violence. I think there needs to be a more comprehensive description of the methods of recruitment. In particular, how were participants recruited without the knowledge of managers? If this wasn't possible, what considerations were made for their safety? How was the study described to the entertainment venue managers?

Results

Table 2: Please describe the comparisons for the p-values. For example, the bottom 2 rows appear to contain only those who answered "yes" to each of these items. Are they compared with the "no" responses? Please provide results for the "no" responses if these are the comparison groups.

Discussion

Line 319-320: This sentence needs to be rewritten.

Page 21: I think this whole page is underscoring how difficult it can be to disentangle the effects of physical/sexual or emotional violence on depressive symptoms. While I think this is what the researchers are explaining, I think it would be useful to make it more explicit - "While we found no independent association between physical or sexual violence and depressive symptoms, the high degree of overlap between physical, sexual and emotional violence may make the identification of an independent association difficult in the current investigation" (or something like this). If the variables were all added at the same time to the model, it is possible that the emotional violence measure is just accounting for the variation in the physical or sexual violence measures also.

Reviewer #2: A well-designed study with an appropriate analysis. The findings are in the expected direction and are consistent with existing literature; however, the study significantly adds to the literature and fills important gaps in the population's understanding of the issue.

Method

According to the methodology, 10 percent of entertainment venues were chosen from each location, and a maximum of seven FEWs were chosen from each entertainment facility. However, it appears that one site, Phnom Penh, has more than 10 percent of the total venues and hence represents a more proportion of FEWs in the study (n = 267, nearly 41 percent). To better understand the variances in the selection process, additional clarity is required on this point.

Analysis

Because the GVB is one of the primary study indicators, analysing GBV by background profile may provide readers with a better understanding.

Results and Discussion

1. The majority of the FSWs were mostly (n=399) and satisfied (n=56) with their current work; only a few were unsatisfied (n=21). Some discussion or reflection would be beneficial for women in abusive workplaces with/without mental health conditions to rate the satisfaction of their current employment.

2. Some thoughts on how current policy fails to protect FEWs from GBV and fails to address their mental health needs.

6. PLOS authors have the option to publish the peer review history of their article (what does this mean?). If published, this will include your full peer review and any attached files.

**Do you want your identity to be public for this peer review?** For information about this choice, including consent withdrawal, please see our Privacy Policy.

Reviewer #1: No

Reviewer #2: **Yes: **Kanougiya Suman

---

## [Editor Report · Decision Letter 1]

14 Jul 2022

Gender-based violence and depressive symptoms among female entertainment workers in Cambodia: a cross-sectional study

PGPH-D-21-00861R1

Dear Dr. Yi,

We are pleased to inform you that your manuscript 'Gender-based violence and depressive symptoms among female entertainment workers in Cambodia: a cross-sectional study' has been provisionally accepted for publication in PLOS Global Public Health.

Best regards,

Muthusamy Sivakami

Academic Editor